# Increasing Aerosol Direct Effect Despite Declining Global Emissions in MPI-ESM1.2

Antoine Hermant[1,2], Linnea Huusko[1], and Thorsten Mauritsen[1]

[1]Department of Meteorology, Stockholm University, Stockholm, Sweden
[2]Climate and Environmental Physics, Physics Institute, University of Bern, Switzerland

**Correspondence:** Antoine Hermant (antoine.hermant@unibe.ch)

**Abstract.** Anthropogenic aerosol particles partially mask global warming driven by greenhouse gases, both directly by reflecting sunlight back to space and indirectly by increasing cloud reflectivity. In recent decades, the emissions of anthropogenic aerosols have declined globally, and at the same time shifted from the North American and European regions to foremost Southeast Asia. Using simulations with the Max Planck Institute Earth System Model version 1.2 (MPI-ESM1.2) we find that the direct effect of aerosols has continued to increase, despite declining emissions. Concurrently, the indirect effect has diminished in approximate proportion to emissions. In this model, which employs parameterized aerosol effects with constant regional direct effect efficiency, the enhanced efficiency of aerosol radiative forcing to emissions is associated with less cloud masking, longer atmospheric residence time, and differences in aerosol optical properties.

## 1 Introduction

The state of Earth's climate is determined by the delicate balance between the incoming solar energy and the energy the Earth reflects and radiates back to space. Greenhouse gases, such as $CO_2$, act to warm the Earth by reducing the radiation emitted to space (radiative forcing), resulting in an accumulation of energy in the climate system and, consequently, an increase in the surface temperature, $T_s$. A linear radiative balance framework can be used to study the response in temperature, $\Delta T_s$, to an applied radiative forcing, $F$, on the net energy balance at the top of the atmosphere, $N$:

$$N = F + \lambda \Delta T_s, \tag{1}$$

where $\lambda$ is the total feedback parameter of the system, which must be negative to yield a stable climate. The feedback parameter can be itself divided into the sum of the individual climate feedbacks, such as water vapour, surface albedo, cloud, and temperature feedbacks. They enhance or dampen the imbalance induced by an applied forcing. Knowledge of how anthropogenic activity affects the radiative balance is crucial for understanding how the climate may change in the future, and currently the largest contributor of uncertainty to estimates of the total anthropogenic forcing is aerosols (Forster et al., 2021).

Aerosols are small particles, emitted by natural and human sources, which affect the radiative balance of the climate system through direct and indirect mechanisms. Directly, they interact with solar radiation by scattering and reflecting it back to space or absorbing it (direct effect). Indirectly, they interact with clouds, for example through the Twomey effect (Twomey, 1974) by increasing the number density of droplets in clouds, making them more reflective to solar radiation (indirect effect).

Due to their short lifetime in the atmosphere, aerosols are heterogeneously distributed in space and time. The heterogeneity, together with interactions between aerosols and clouds, makes the aerosol forcing difficult to constrain, and thus adds uncertainty to estimates of the total radiative imbalance of the climate system. This uncertainty poses challenges in assessing how aerosols may mask the effects of global warming caused by greenhouse gases and the trajectory of future global warming. The situation is made even more complex by the shift in dominant aerosol emission sources from Europe and North America to Southeast Asian regions that has occurred as a result of air quality regulations in the last decades (Smith et al., 2011).

It is therefore a community priority to narrow down the uncertainty in aerosol forcing (Bellouin et al., 2020). Multiple approaches to this end have been taken, including process understanding usually in the form of global models (e.g. Fiedler et al., 2023), satellite observational constraints based on internal variability or volcanic eruptions (e.g. Gryspeerdt et al., 2017; McCoy et al., 2017; Malavelle et al., 2017), or top-down approaches based on the observed global warming (Stevens, 2015; Kretzschmar et al., 2017; Booth et al., 2018). In particular for top-down constraints it is essential to relate the historical evolution of aerosol emissions to radiative forcing.

Therefore, in the current study we separate the historical evolution of the direct and indirect aerosol forcing in a climate model, and relate these to global aerosol emissions. The effect of the regional redistribution in recent decades is investigated, and the underlying mechanisms elucidated.

## 2 Method

### 2.1 MPI-ESM1.2 and MACv2-SP

We study the historical evolution of the aerosol forcing using the state-of-the-art global climate model Max Planck Institute for Meteorology Earth System Model version 1.2. MPI-ESM1.2 (Mauritsen et al., 2019). MPI-ESM1.2 participated in the Coupled Model Intercomparison Project Phase 6 (CMIP6) and successfully reproduces the observed warming from pre-industrial levels (Mauritsen and Roeckner, 2020). In this study we use the coarse resolution version of the model (CR, see Mauritsen et al. (2019)) to simulate the historical period from 1850 to present day, with input as in the CMIP6 historical scenario.

The radiative transfer scheme of MPI-ESM1.2 uses the Simple Plumes implementation of the second version of the Max Planck Institute Aerosol Climatology (MACv2-SP) to represent the aerosol impact on the radiation (Stevens et al., 2017). MACv2-SP provides a parameterization of optical properties of anthropogenic aerosols and the resulting Twomey effect (Twomey, 1974; Stevens et al., 2017). It has been designed to provide a uniform and easily controlled representation of anthropogenic aerosol perturbations for the CMIP6 framework (Eyring et al., 2016; Pincus et al., 2016). In the model, aerosol emissions are represented by nine spatial plumes that are associated with emissions from major anthropogenic source regions (Stevens et al., 2017).

The prescribed aerosol optical properties are based on ground-based sun-photometer measurements provided by the Max Planck Institute Aerosol Climatology, MAC (Kinne et al., 2013), and merged onto global model background maps from AeroCom for the present-day (2005) distribution of mid-visible anthropogenic aerosol optical depth (AOD). AOD at 440, 500 and 870 nm as well as absorbing AOD at 550 nm are considered, along with additional contributions of coarse and fine-mode

aerosol particles. To represent forcing variations from pre-industrial (1850) to 2016, MACv2-SP uses emission inventory estimates of $SO_2$ and $NH_3$ from the Community Emissions Data System (CEDS). These two aerosol precursors are weighted to calculate the plume scale factor (Stevens et al., 2017). This weighting of different aerosol emissions gives rise to a global emission unit of Tg of $SO_2$ equivalent. We use this unit throughout our analysis. However, the effects of changes in aerosol composition over time within plumes are not considered. For instance, the 'brightening' of aerosol composition due to reduction in BC fraction relative to $SO_2$ prior to 1970 and the opposite trend afterwards is not explicitly included in MACv2-SP (Stevens et al., 2017).

Two types of emissions are considered: industrial emissions for the plumes in Europe, North America, Australia, East and South Asia and biomass burning emissions for the plumes in South America, the Maritime Continent, North and South Central Africa. These two types of emissions differ in their seasonal cycle amplitude, single-scattering albedo (SSA), and the strength of the Twomey effect. This distinction is essential to reflect the composition of specific aerosol species in each plume type. The measurements suggest a SSA of 0.93 for industrial areas and 0.87 for regions dominated by seasonal biomass burning (Stevens et al., 2017). This difference accounts for the greater absorption of biomass burning aerosols, and therefore for some of the global aerosol composition changes over time, even though these changes are not included within each plume.

To model the Twomey effect, Stevens et al. (2017) used satellite observations to derive a relationship between the cloud droplet number density and the fine-mode AOD. With this representation, anthropogenic aerosols cause a greater increase in cloud optical thickness when the atmospheric environment is initially pristine in terms of aerosols. A complete description of MACv2-SP can be found in Stevens et al. (2017).

There are limitations in the MAC-SP representation of aerosol forcing, as it only takes into account two main aerosol precursors and does not include changes in aerosol compositions over time or interactions with the atmosphere. Nevertheless, Stevens (2015) argues that this representation successfully captures the main features seen in more complex models, both in terms of global signal and regional patterns. We consider these limitations throughout the study and discuss the results in that context.

## 2.2 Radiative Forcing Calculation

In the literature, various metrics have been proposed to assess the impact of a radiative perturbation on Earth's energy balance. When a perturbation is introduced into the climate system, rapid adjustments occur due to rapid stratospheric temperature change. Radiative forcing (RF) is used to quantify the radiative imbalance resulting from an applied perturbation, taking these rapid adjustments into account. The effective radiative forcing (ERF) includes further adjustments of the system, accounting for all tropospheric and land surface adjustments.

We use the Partial Radiative Perturbation (PRP) method to calculate the radiative forcing from aerosols in the MPI-ESM1.2 simulations. In this method the radiative effect of a change in a certain state variable, for example the effect of a change in surface albedo between a control and perturbed climate run, is calculated as the difference between two radiation calls. The method was first described by Wetherald and Manabe (1988) and Colman and McAvaney (1997), and implemented in MPI-ESM by Meraner et al. (2013) and Block and Mauritsen (2013) as a two-sided version (including the backward perturbation

calculation in 2 extra calls), as described by Klocke et al. (2013). Another version of the PRP diagnostic has also been used by Mülmenstädt et al. (2019) to assess the radiative effects of aerosols in a different version of the atmospheric model component of MPI-ESM1.2.

The PRP method was designed to evaluate the respective contributions of individual forcings and feedbacks on the radiative imbalance (Eq. 1). Here, we integrated the anthropogenic aerosol perturbation provided by MACv2-SP into the PRP module of MPI-ESM1.2. This enabled us to estimate the instantaneous radiative forcing from anthropogenic aerosols independently of climate feedbacks and atmospheric adjustments. Furthermore, as MACv2-SP provides two distinct prescribed perturbations for the direct and indirect effects of aerosols, we can substitute them one at a time into the PRP method to evaluate their

respective contributions. This approach allows us to conduct regular historical simulations in MPI-ESM1.2 and investigate the past evolution of anthropogenic aerosol effects on the climate system.

## 3   Results

In the following section we present the simulated anthropogenic aerosol forcing throughout the historical period. We separate the contributions from the direct and indirect effect and show how the forcing varies depending on the location of the aerosol

emissions. Finally, we investigate the mechanisms governing regional differences in the relationship between aerosol emissions and radiative forcing.

### 3.1   Historical Aerosol Forcing

In the historical simulation, the PRP diagnostic reveals an increasingly negative forcing from aerosols throughout the century, despite the global reduction in aerosol emissions in recent decades, as shown in Figure 1. The persistently negative trend in

the total aerosol forcing is primarily driven by the direct effect, which continues to increase even after the implementation of air quality regulations in Europe and North America since the 1970s and 80s. Meanwhile, the indirect effect is reduced in approximate proportion to the decreasing emissions.

In addition to the global decrease in aerosol emissions, the period spanning from the 1970s to 2005 witnessed a shift in aerosol emission patterns. Early in the historical period global aerosol emissions were dominated by emissions from Europe

and North America. In recent decades, South and East Asia have become the dominant source regions. The subsequent sections investigate the role played by this geographical shift in explaining the observed inconsistency between global aerosol emissions and forcing.

### 3.2   Forcing from Regional Aerosol Sources

As detailed in Section 2, MACv2-SP provides a parameterization for anthropogenic aerosols, incorporating nine distinct plumes

that represent various anthropogenic emission regions. To assess the aerosol forcing from each of these regions throughout the historical period, we substituted one plume at a time into the PRP calculation. Figure 2 shows the resulting forcing values from each region against the corresponding aerosol emissions in Tg of $SO_2$ equivalent. Regressing the induced forcing against

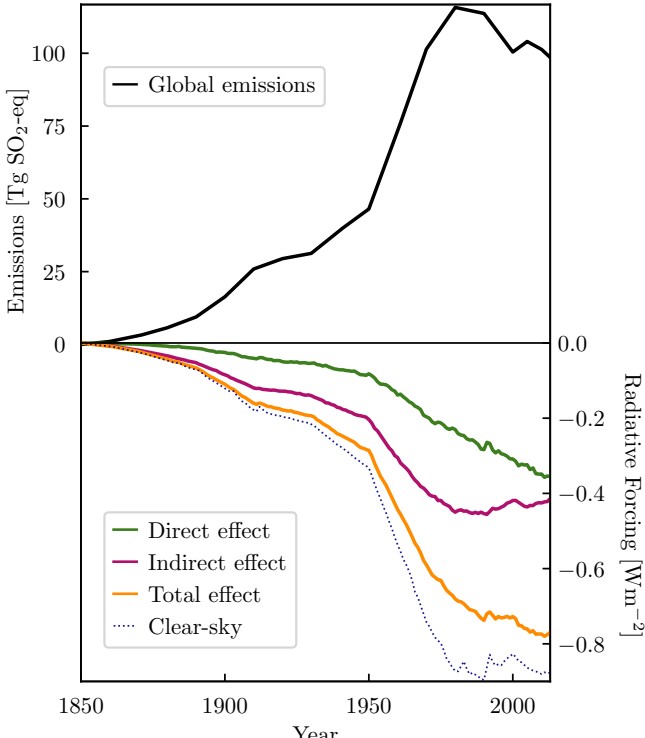

**Figure 1.** Historical forcing from anthropogenic aerosols. The top part shows the historical global emissions of aerosols and the bottom part shows the induced radiative forcing in MPI-ESM1.2. Values are global yearly means.

the associated emission level, we obtain a value of the regional aerosol efficiency in $Wm^{-2}$ per Tg of $SO_2$ equivalent. For all regressions shown in Figure 2, the coefficients of determination $R^2$ fall within 89-99% confidence. This implies that the
calculated efficiencies are consistent with emission levels and thus with time.

Using our PRP method, we calculate the relationship between forcing and emissions for each individual plume. Significant trends in forcing as they relate to emissions are observed (Figure 2). Quaas et al. (2022) demonstrate the relationship between clear-sky aerosol ERF trends and trends in sulfate precursors, noting significant declines in major source regions from North America, Europe and East Asia, alongside increases in India and surrounding regions (which we refer to as South Asia). These
findings align with our results across the four major industrial regions. Furthermore, Quaas et al. (2022) support the idea that trends in sulfate precursors are driving aerosol forcing increases, which is consistent with our results as the aerosol forcing in MACv2-SP is predominantly driven by sulfate emissions (Stevens et al., 2017).

In Figure 2a, we observe a significant variability in efficiency of the direct effect among major industrial regions, such as Europe, North America, and East and South Asia. Notably, South Asia exhibits an efficiency 20 times greater than Europe,
representing the most substantial difference in efficiency across these regions. On the other hand, Figure 2b shows a relatively more consistent relationship between forcing and emissions across regions for the indirect effect.

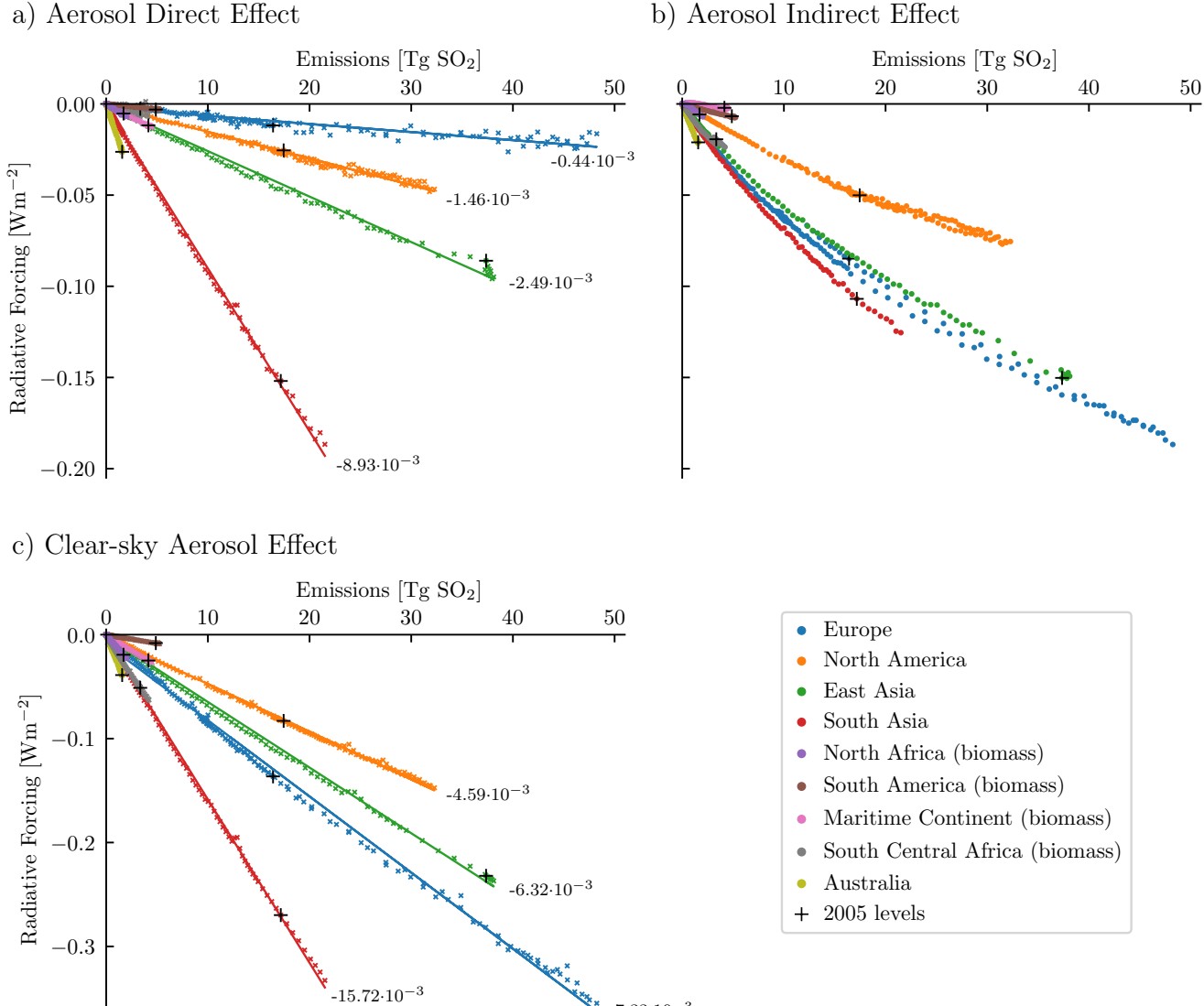

**Figure 2.** Aerosol forcing from individual emission regions against regional emission levels. The values are global yearly means for each year from 1850 to 2013 separated into the direct (a) and indirect (b) effect. Panel c) shows the clear-sky aerosol effect. The numbers annotated in panels a) and c) are the efficiencies in $Wm^{-2}$ per Tg of $SO_2$ equivalent for the major emission regions, based on linear regression (89 to 99% confidence). Black crosses indicate the emission levels and resulting radiative forcing for the individual sources at the reference year of 2005.

The variation in regional aerosol efficiency explains the persistent increase in the global direct effect despite decreasing emissions. Regions with higher efficiencies have a more substantial impact on the global direct effect despite emitting fewer aerosols. This effect becomes particularly important when considering the shift in aerosol patterns from 1980 to 2005. During this period, aerosol emissions shifted from Europe and North America to South and East Asia, where the efficiency is higher. Consequently, despite reduced global emissions during this period, the global aerosol forcing continued to increase. It is important to note that the discrepancy in aerosol efficiencies across emission regions is observed for the year 2005 where the present-day aerosol climatology (see Section 2.1) is directly applied. Subsequent sections delve into the mechanisms that underlie these modeled regional variations in efficiency.

## 3.3 All-sky and Clear-sky Aerosol Forcing

We examine the outcomes of the PRP performed under both all-sky and clear-sky conditions. The results show that under all-sky conditions the direct effect primarily causes a radiative forcing in the vicinity of emission sources (see Figure 3c). In contrast, the indirect effect is more pronounced over remote regions (see Figure 3b) and is larger than the direct effect on global average. These results are in line with Huusko et al. (2022), who used a different method for estimating the spatial patterns of the aerosol direct and indirect effects in MPI-ESM1.2.

The clear-sky global aerosol forcing surpasses the all-sky global total aerosol forcing (including direct and indirect effects, see Figure 1). It is essential to note that under clear-sky conditions only the direct effect applies. Interestingly, the clear-sky aerosol forcing is more than twice as large as the direct effect observed in all-sky conditions. This pattern remains consistent across all emission regions, with clear-sky aerosol forcing consistently exceeding the all-sky direct effect (see Figure 2a and c). Under all-sky conditions, the presence of extensive cloud cover locally results in positive forcing from the direct effect of aerosols (see Figure 3c and e), as the presence of clouds below aerosols moderates the net effect of aerosol scattering while amplifying the net effect of aerosol absorption (Li et al., 2022; Bellouin et al., 2020). With single-scattering albedo of 0.93 and 0.87 for industrial and biomass burning emissions, respectively, in the model (Stevens et al., 2017), absorption prevails in the presence of clouds, resulting in positive direct effect of aerosols.

In addition, in regions with persistent cloud systems, the negative forcing from the indirect effect and the positive direct effect tend to balance each other (see Figure 3a and e). This mechanism has significant implications for regional emission efficiency. In particular, it explains why Europe, which exhibits a weak efficiency under all-sky conditions (Figure 2a) due to a positive direct effect at high latitudes (Figure 3), demonstrates greater efficiency under clear-sky conditions (Figure 2c). Looking at clear-sky conditions significantly narrows the gap in regional efficiencies. In South Asia, the efficiency in clear-sky conditions is only 2.1 times greater than in Europe, which is significantly lower than the factor 20 difference observed in all-sky direct effect. The most pronounced difference under clear-sky conditions is seen between South Asia and North America, with South Asia showing a efficiency 3.2 times greater than North America.

The effect of cloud cover on the direct effect of anthropogenic aerosols emerges as the main factor influencing regional efficiency. This largely explains the consistent increase in negative aerosol forcing despite reduced emissions in the last decades. As the modeled cloud cover is less persistent at the emission sources in South and East Asian regions compared to Europe,

the shift in aerosol patterns results in an enhanced global direct effect. The indirect effect instead induces a forcing in remote regions downstream of emissions. The strength of this effect globally follows the global emissions despite the shift in the emission pattern. The increase in aerosol efficiency resulting from the shift in aerosol spatial pattern is critical to explaining the increase in aerosol forcing between the mid-1970s and the mid-2000s, despite similar emissions levels. However, the disparity in emission efficiencies between regions remains substantial even under clear-sky conditions. The following sections investigate other factors contributing to these regional differences.

### 3.4 MACv2-SP Aerosol Representation and Regional Variation

The MACv2-SP has been designed to simplify the representation of anthropogenic aerosols in climate models through a straightforward parameterization. It provides monthly mean Aerosol Optical Depth (AOD) values based on ground-based measurements, representing the 2005 spatial distribution. To get the spatial pattern in other years the 2005 pattern is scaled with estimates of historical emissions (see Section 2.1 and Stevens et al. (2017)). The measured AOD values are influenced by the rate of aerosol removal at a given location, so even though aerosol interactions with the atmosphere are not represented in MACv2-SP, removal processes may be implicitly recorded. In consequence, the AOD values in the model may not always be directly proportional to regional emission levels. For example, wet deposition is the dominant sink of sulfate aerosols from industrial sources (Textor et al., 2006), and this deposition mechanism is particularly prominent over the eastern coasts of North America and East Asia (Rodhe et al., 2002), causing a relatively small AOD per unit of emissions there.

Figure 4a shows the clear-sky aerosol forcing against the corresponding AOD for each region. This representation noticeably reduces the difference in aerosol efficiency between regions compared to Figure 2c, indicating that AOD is a better predictor of aerosol forcing than emissions. For instance, the efficiency of South Asia is 1.3 times greater than that of Europe when measured in $Wm^{-2}$ per unit of optical depth, whereas it was 2.1 times greater when measured in $Wm^{-2}$ per unit of emissions (in Tg of $SO_2$-eq). Interestingly, when considering AOD levels, both Asian regions exhibit similar efficiencies, which is not the case when considering emissions. A possible explanation for this is that strong wet deposition in East Asia contributes to the removal of aerosols (Rodhe et al., 2002), resulting in a lower AOD per unit of emissions and thus a weaker direct effect in this region. Conversely, South Asia exhibits weaker wet deposition (Rodhe et al., 2002), allowing for greater forcing per unit of emissions.

This difference in aerosol removal patterns is the second most important explanation for the continued increase in aerosol forcing despite reduced emissions. The shift in aerosol patterns from Europe and North America towards Southeast Asian regions, with weaker wet deposition, prolongs the residence time of aerosols in the atmosphere, consequently enhancing the aerosol efficiency. In the last section, we suggest additional explanations for the remaining minor differences in aerosol efficiency between regions.

### 3.5 Aerosol Single-Scattering and Surface Albedo

The distinction between industrial and biomass aerosol emissions affects primarily the Single-Scattering Albedo (SSA) parameter provided by MACv2-SP, which is 0.93 for industrial and 0.87 for biomass burning (Stevens et al., 2017). This distinction is

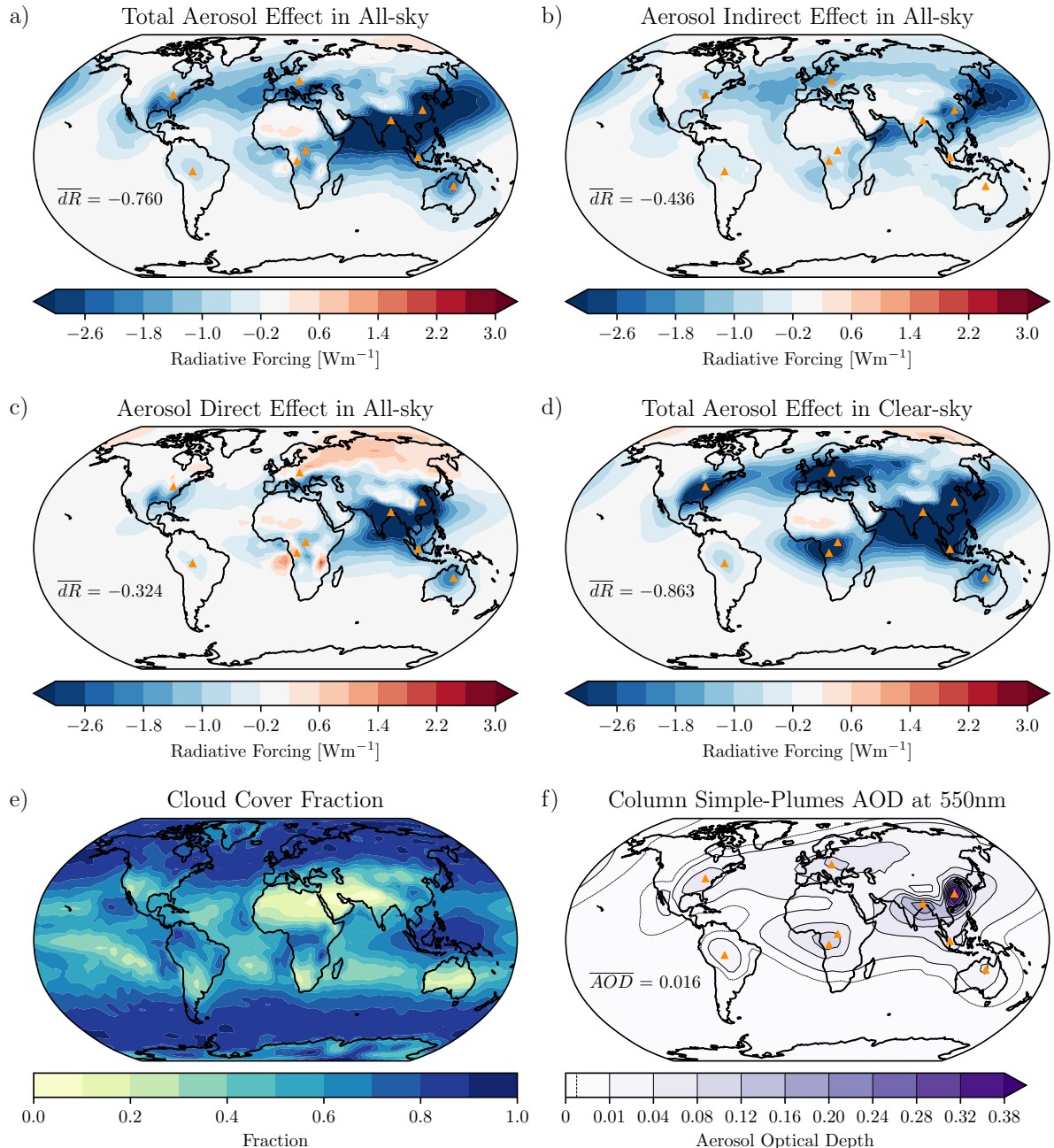

**Figure 3.** Present-day (2005) spatial pattern (yearly mean) of a) the total aerosol forcing, separated into b) the indirect effect and c) the direct effect; d) the clear-sky aerosol effect; e) the cloud cover fraction. Panel f) shows the Column Aerosol Optical Depth at 550nm from the MACv2-SP parameterization, with dashed-line showing low AOD value contour (0.0025). Values annotated on the maps are global means.

## a) Clear-sky Aerosol Forcing

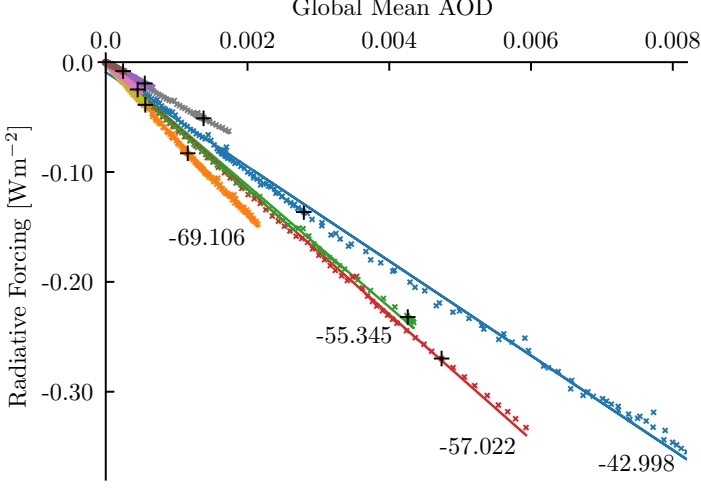

## b) Clear-sky Aerosol Forcing with Equal SSA

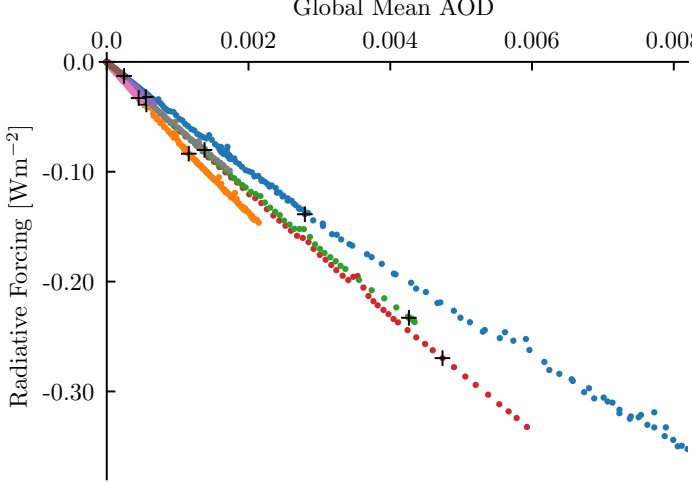

**Figure 4.** Aerosol forcing from individual emissions regions against regional column aerosol optical depth. Plots are global yearly means for each year from 1850 to 2013 with panel a) showing the clear-sky aerosol effect. Panel b) shows the same but in a new experiment in which the single-scattering albedo was set to the same value for all sources. Values annotated on the plots are the efficiencies in $Wm^{-2}$ per unit of optical depth of the major emission regions, based on linear regression. Black crosses indicate the emission levels and resulting radiative forcing for the individual sources at the reference year of 2005.

made to account for the important differences in properties between industrial and biomass burning aerosols and to fit the 2005 aerosol climatology (see Section 2.1 and Stevens et al. (2017)). We observe that the greater shortwave absorption by biomass burning aerosols results in weaker efficiencies when compared to industrial regions. Figure 4b is similar to Figure 4a but with data from a new simulation where the SSA was set to the same value (0.93) for all sources, substantially reducing the spread for the biomass burning source regions. This demonstrates the influence of aerosol properties on the aerosol efficiency and thus the influence of the aerosol climatology used. It is important to note that the influence of SSA holds true in all-sky conditions as well, as SSA defines the ratio of scattering efficiency to total extinction efficiency. However, it plays a relatively minor role in the total global mean discrepancy between emissions and aerosol forcing, given that biomass burning regions have relatively smaller emissions and forcing compared to industrial sources.

It is important to note that despite this distinction in aerosol composition between industrial and biomass burning plumes, changes in aerosol composition over time is not represented in MACv2-SP (see Section 2.1). It can be argued that this may have implications on the evolution of the aerosol efficiency over time in this study. For example, as mentioned in Section 2.1, the increase in black carbon fraction after 1970 should contribute to a decrease in aerosol efficiency, which is contradictory with our results. However, this increase in black carbon fraction is accompanied by a decrease in emissions from European and North American industrial plumes, and increase in emissions from Asian industrial plumes. Our results indicate that the increase in aerosol efficiency is primarily driven by South Asian regions, which exhibit greater efficiency compared to other regions. Importantly, South Asia's aerosol composition has remained consistent since 1980 (Stevens et al., 2017), suggesting that its efficiency has likely remained stable over recent decades. In our study, South Asia has been the primary driver of aerosol efficiency since the onset of the global decline in aerosol emissions. Given this region's dominant role and the consistency of its aerosol composition, we conclude that incorporating temporal changes in aerosol composition would have minimal impact on our findings. Consequently, such changes are unlikely to explain the observed global discrepancy between global aerosol emissions and the total direct effect.

Some spread still remains between the regions, suggesting that some other mechanisms also influence the efficiency. For example, the influence of anthropogenic aerosols on the radiative balance depends on the nature of the underlying surface (Li et al., 2022), suggesting that the remaining differences in clear-sky efficiency among emission regions may be partly associated with differences in surface albedo between the regions.

## 4 Conclusions

We have investigated the relationship between aerosol emissions and radiative forcing in the global climate model MPI-ESM1.2. Our results reveal an increase in global mean aerosol radiative forcing in recent decades, despite a global reduction in aerosol emissions. This increase is driven primarily by the direct effect, while the indirect effect remains more consistent with emission levels. The increase in the direct effect is associated with regional shifts in emissions. Historically, Europe and North America have been the primary sources of aerosol emissions, but since the 1970s emissions have shifted to South and East Asia, where the radiative forcing per unit of emissions is larger.

The primary mechanism driving the disparity between global aerosol emissions and radiative forcing is the masking effect of clouds. Mid- to high-latitude regions, characterised by substantial cloud cover, exhibit enhanced aerosol absorption relative to scattering resulting in a weaker negative or even positive direct effect. The recent shift of aerosol emissions to South and East Asia, where the cloud cover is less extensive, has led to a more negative global mean aerosol direct effect.

Other significant contributors include the regional variation in aerosol residence time within the atmosphere. Atmospheric conditions in South Asia with weaker removal processes, such as wet deposition, as compared to North America, allow aerosol particles to stay in the atmosphere longer, causing a greater efficiency in terms of radiative forcing per unit of emissions. Furthermore, the optical properties of the aerosol particles themselves can influence the forcing efficiency per unit of emissions. In recent decades, emerging biomass burning source regions have led to the emission of greater quantities of absorbing aerosols, which act to dampen the negative direct effect. Nevertheless, since the biomass burning emissions are relatively small compared to those from industrial sources, this has not offset the global negative increase in the direct effect. Furthermore, although the MACv2-SP parametrization does not include changes in aerosol composition over time, the greater efficiency and stable aerosol composition of the emerging South Asian region suggest that these changes are unlikely to influence the discrepancy observed in our findings. However, this has to be confirmed by similar studies in more explicit aerosol-climate models where changes in aerosol composition over time are included.

When compared to other global climate models, MPI-ESM1.2 has a relatively weak total aerosol forcing (Mauritsen et al., 2019), and Fiedler et al. (2023) have found that the range of aerosol forcing among CMIP6 models is primarily influenced by the strength of the indirect effect. Our findings imply that models with a weaker indirect effect relative to the direct effect may exhibit a radiative forcing less consistent with global emissions, while models with a strong indirect effect are likely to have greater consistency between their aerosol forcing and emissions. This can help explaining the variety in the evolution of the aerosol effective radiative forcing throughout the historical period in CMIP6 observed by Fiedler et al. (2023). The results of this study show that in an ESM with a reasonable representation of the historical evolution aerosol forcing it is possible to have a trend in the direct radiative effect from aerosols that diverges from the trend of emissions. Based on the physical mechanisms explained here we argue that this effect may be realistic. However, we encourage similar studies to verify whether this continuous increase in the direct effect can be found also in other models.

*Code and data availability.* The source code for MPI-ESM1.2 can be accessed via https://mpimet.mpg.de/en/science/models/mpi-esm (Mauritsen et al., 2019). Additionally, the specific parts of the code that were modified and developed for this study, as well as the model outputs and Python scripts used in producing the figures presented in this paper, are accessible through Zenodo at https://doi.org/10.5281/zenodo.10161509 (Hermant et al., 2023).

*Author contributions.* AH conducted the simulations and analysis, and wrote most of the paper. All authors contributed to the research and writing.

*Competing interests.* The authors declare that they have no conflict of interest.

*Acknowledgements.* The computational resources were provided by the National Academic Infrastructure for Supercomputing in Sweden (NAISS) partially funded by the Swedish Research Council through grant agreements no. 2022-06725. This project was funded by the Swedish e-Science Research Center (SeRC), the European Research Council (ERC) (grant agreement no. 770765), the European Union's Horizon 2020 research and innovation program (grant agreement nos. 820829, 101003470), and the Swedish Research Council (VR) (grant agreement no. 2022-03262).

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
