# Peer review of "Increasing Aerosol Direct Effect Despite Declining Global Emissions in MPI-ESM1.2"

_EGUsphere, 2024_

## Author Comment (AC2)

**Reply to reviewer 2**

Thank you for your insightful comments on our manuscript. We appreciate your attention to detail and would like to address the concerns you raised regarding the radiative forcing efficiency within the MACv2-SP aerosol prescription. We acknowledge the critical role of parametrisation choices in determining aerosol radiative forcing as well as the limitations of this method, and understand your point about the inherent influence of these choices. It is important to note that MACv2-SP was specifically developed to capture aerosol radiative effects on climate using ground-based and satellite measurements as well as emission inventories of key precursors such as SO2, NH3, and BC. In addition, MACv2-SP has been designed to be lightweight and easy to use in different climate models in the CMIP6 framework. Furthermore, Stevens [2015] argues that this representation successfully replicates key features observed in more complex models, both globally and regionally. To address your comments effectively, we will provide additional details on the construction of MACv2-SP to offer further clarity on our analysis. We also recognize the importance of a thorough understanding of Stevens et al. [2017] for contextualizing our work, particularly Section 4: "Time-varying forcing".

**Answers to main comments**

1. We understand your concern about the strength of the South Asian plume. MACv2-SP uses CEDS emission inventories for scaling the forcing, including SO2, NH3 and BC for all plumes. Both reflective and absorbing aerosols are considered. MACv2-SP has been designed to fit climatological values of aerosol properties at the year 2005. "High-quality data by ground-based sun-photometer" are merged onto "global model background maps from AeroCom" [Stevens et al., 2017]. Different aerosol properties are considered: AOD at 440, 550 and 870 nm, absorbing AOD at 550 nm and coarse and fine-mode aerosol particles [Stevens et al., 2017]. Using this aerosol climatology retrieved from direct measurements, our calculation indicate a strong direct effect efficiency from South Asia, both against emissions and AOD in 2005 (see Response Table 1). Since that the year 2005 applies aerosol climatology from measurements and AeroCom, we are uncertain why South Asian aerosols would be expected to be strongly absorbing.

   Furthermore, there must have been a misunderstanding about Section 3.5. We do suggest that SSA significantly drives direct effect forcing efficiency (line 174-175: "a new simulation where the SSA was set to the same value for all sources, substantially reducing the spread"). SSA does have an influence on the forcing, which explains the difference between plumes dominated by industrial aerosols and plumes dominated by biomass burning aerosols. But we emphasize that since the recent increases in biomass burning in the Southern Hemisphere have a minor contribution to the total aerosol direct effect, they have a minimal impact on the global forcing-to-emission decoupling.

2. Emission rates are used to scale the 2005 aerosol climatology derived from measurements and AerCom [Stevens et al., 2017]. Analysis of the year 2005, where the measured climatology is applied (without any scaling), already suggests a strong regional dependence of the direct effect (see Response Table 1. Figure 2.c suggests that the difference between regions is in large part explain by cloud masking. Furthermore, Figure 4.a shows that the spread is reduced when considering AOD instead of emissions. To explain this, we do speculate that some aerosol processes are implicitly recorded in the instrumental measurements, such as aerosol removal. We do not completely understand your concern about the influence of residence times on radiative forcing, since we observe this discrepancy between emissions and AOD in 2005. Longer residence time results in greater measured AOD per unit of emission, resulting in greater forcing efficiency per unit of emission. Since the forcing is scaled with emissions inventories, the discrepancy is spread over time. Response Table 1 highlights the emissions and forcing values for each plume in the year 2005. The efficiencies in the Table are directly calculated from 2005 for comparison with the ones calculated in the manuscript figures. While in the manuscript we calculated the efficiencies via linear regression both against emissions and AODs, we acknowledge that we should more explicitly state that these efficiencies are consistent with emissions and thus time. We intent to clarify this in the manuscript.

**Other comments**

1. Lines 21-23: Both direct and indirect effect and aerosol-radiation (ari) and aerosol-cloud interactions (aci) appear in the literature, and we find that the direct and indirect effect terminology are more intuitive and connect better with the framework of radiative forcing as an instantaneous effect on the radiative balance and subsequent interactions and adjustments within the climate system.

| Source region | Emissions | AOD | ADE | ADE/E | ADE/AOD |
|---|---|---|---|---|---|
| Europe | 16.41 | 2.79 | -0.012 | -0.72 | -4.22 |
| North America | 17.45 | 1.16 | -0.025 | -1.46 | -22.0 |
| East Asia | 37.36 | 4.26 | -0.086 | -2.30 | -20.18 |
| South Asia | 17.17 | 4.74 | -0.152 | -8.85 | -32.06 |
| North Asia | 1.70 | 0.55 | -0.005 | -3.09 | -9.54 |
| North Africa | 4.88 | 0.24 | -0.003 | -0.63 | -12.64 |
| South America | 4.15 | 0.45 | -0.012 | -2.83 | -26.07 |
| Maritime Continent | 3.35 | 1.38 | -0.003 | -0.80 | -1.94 |
| Australia | 1.57 | 0.56 | -0.026 | -16.73 | -47.01 |

Response Table 1: Aerosol direct effect efficiencies per source region in 2005. Emissions are Equivalent SO2 in Tg SO2, AOD [$10^{-3}$], Aerosol Direct Effect (ADE) in [$Wm^{-2}$], ADE/E in [$10^{-3}$ $Wm^{-2}$] per emission unit, ADE/AOD in [$Wm^{-2}$] per AOD unit

2. Line 29: We appreciate that you highlight the relevance of Quaas et al. [2022]. Section 5 of their work emphasizes the close relationship between clear-sky solar ERFaer trends and trends in sulfate precursors, noting significant declines in major source regions from North America, Europe, and East Asia, alongside increases in India and surrounding regions. These findings align with our results across the 4 major industrial plumes and address your concerns regarding the strength of South Asia and the dominance of sulfate in MACv2-SP. Notably, their study supports our results by indicating that trends in sulfate precursors are driving aerosol forcing increases in regions like 'India and surrounding areas' (referred to as South Asia in MACv2-SP) over recent decades. We intend to include this comparison into the manuscript in Section 3.2.

3. Line 73: We systematically employ the two-sided PRP method described in Klocke et al. [2013].To keep our text concise, we opted not to delve into a detailed explanation of the PRP methodology in this paper. The original forward PRP (Wetherald and Manabe [1988]) consists in sequentially substituting specific state variable fields from the current climate state into a reference state, while keeping all other variables constant at the reference level. This partial perturbation approach allows for assessing each variable's contribution to the total radiative forcing. However, the perturbation is sensitive to the state in which it is introduced [Colman and McAvaney, 1997] and de-correlating fields when substituting them introduces unintended perturbations [Klocke et al., 2013]. The proposed method to partially address these approximations is to apply the partial radiative perturbation forward and backward. The backward perturbation consists in substituting the variable fields from the reference state into the current state. By averaging the results of forward and backward computations (two-sided approach), a more accurate estimation of forcing is achieved. We judge that a detailed description of this methodology would be too technical for the scope of this paper, thus we refer the reader to Klocke et al. [2013] and Colman and McAvaney [1997] for a more comprehensive description.

4. Line 103: As mentioned earlier, emissions in Tg of SO2 equivalent are calculated taking into account SO2, NH3 and BC emission inventories from CEDS. We acknowledge that we should more explicitly highlight this in the paper and intent to do so in the revised version, since we use SO2 equivalent as a unit in our main results. For a full description of SO2 equivalent calculation, the reader can refer to Stevens et al. [2017].

5. Line 128: You are right that absorption prevails in the presence of clouds only if the aerosols are above clouds. We will mention this in section 3.3.

6. We use the MACv2-SP version from Stevens et al. [2017] that integrates the CEDS emission estimates used for CMIP6 historical forcing input data. As mentioned previously, the MACv2-SP reference year 2005 uses instrumental measurements of the aerosol climatology. Since our results for the year 2005 already suggest a regional dependence of the aerosol efficiency and imply the decoupling, we do not think that more recent emission estimates would affect the results. The emission estimates are only used to scale the 2005 reference to represent the time-varying forcing. In the context of our study, this helps to clearly observe the decoupling in Figure 1, but our key results are the spatial representation with Figure 2,3 and 4.

Additionally, the results from Quaas et al. [2022] uses these 2021 updated CEDS emission estimates. Using this new dataset they obtain the results discussed in our answer to your comment on Line 29.

Since their results are consistent with our results, we believe that this update would have little effect on our results, and is unlikely to impact the outcomes of our research.

In summary, here is what we intend to include in the revised version of the manuscript:

- Scatter points on Figure 2 and 4 to highlight 2005 values (as presented here in Response Table 1), as well as details on the temporal-consistency of the plume efficiencies.

- A more exhaustive description of the MACv2-SP aerosol representation, especially summarizing Section 4 on Time-varying forcing in Stevens et al. [2017] (as formulated in this answer to reviewer 2).

- Clarity on the effect of SSA and on the SO2-equivalent emission unit.

**References**

Bjorn Stevens. Rethinking the lower bound on aerosol radiative forcing. *Journal of Climate*, 28(12):4794–4819, jun 2015. doi: 10.1175/jcli-d-14-00656.1.

Bjorn Stevens, Stephanie Fiedler, Stefan Kinne, Karsten Peters, Sebastian Rast, Jobst Müsse, Steven J. Smith, and Thorsten Mauritsen. MACv2-SP: a parameterization of anthropogenic aerosol optical properties and an associated Twomey effect for use in CMIP6. *Geoscientific Model Development*, 2017. doi: 10.5194/gmd-10-433-2017.

Johannes Quaas, Hailing Jia, Chris Smith, Anna Lea Albright, Wenche Aas, Nicolas Bellouin, Olivier Boucher, Marie Doutriaux-Boucher, Piers M. Forster, Daniel Grosvenor, Stuart Jenkins, Zbigniew Klimont, Norman G. Loeb, Xiaoyan Ma, Vaishali Naik, Fabien Paulot, Philip Stier, Martin Wild, Gunnar Myhre, and Michael Schulz. Robust evidence for reversal of the trend in aerosol effective climate forcing. *Atmospheric Chemistry and Physics*, 22(18):12221–12239, September 2022. ISSN 1680-7324. doi: 10.5194/acp-22-12221-2022.

Daniel Klocke, Johannes Quaas, and Bjorn Stevens. Assessment of different metrics for physical climate feedbacks. *Climate Dynamics*, 2013. doi: 10.1007/s00382-013-1757-1.

R. T. Wetherald and S. Manabe. Cloud Feedback Processes in a General Circulation Model. *Journal of the Atmospheric Sciences*, 1988. doi: 10.1175/1520-0469(1988)045⟨1397:cfpiag⟩2.0.co;2.

R. A. Colman and B. J. McAvaney. A study of general circulation model climate feedbacks determined from perturbed sea surface temperature experiments. *Journal of Geophysical Research: Atmospheres*, 1997. doi: 10.1029/97jd00206.

---

## Author Comment (AC3)

**Reply to reviewer 1**

Thank you for your valuable comments. This study indeed aims to explore the potential decoupling between aerosol emissions and their direct effects. We acknowledge your concerns regarding the modelling approach employed to estimate historical aerosol forcing in MPI-ESM1.2 through the use of the MACv2-SP parametrization. In this response, we aim to provide clarity on our methodology by providing a description of the MACv2-SP parametrisation and specific results from our study.

1. MACv2-SP combines ground-based measurements of a 2005 aerosol climatology with emission estimates to represent historical changes in aerosol direct and indirect effects. Emissions estimates of SO2, NH3 as well as BC (CEDS) are accounted for in the scaling of the 2005 climatology (See Stevens et al. [2017]: Section 4: 'Time-varying forcing', Table 5,6, Figure 9,10). By weighting the radiative properties of these three compounds with their respective emission estimates, the changes in aerosol properties in time and space are represented. We acknowledge the limitation of this representation, which only takes into account these three (major) species and does not include interactive processes with the atmosphere. Nevertheless, Stevens [2015] argues that this representation of emissions successfully capture the main features seen in more complex models, both in terms of global signal and regional patterns. We intend to clarify and elaborate the representation of the time-varying forcing in the manuscript. While aerosol removal processes are not explicitly represented in MPI-ESM1.2 with MACv2-SP, they are recorded in the in-situ measurements and thus included implicitly in the MACv2-SP representation, and we argue that this provides a sufficient representation of aerosol forcing in the context of this study.

    While we acknowledge your concern regarding the time-variation of aerosol forcing, we intend to provide specific results for the reference year of 2005, since it directly applies the measured aerosol climatology. Utilizing the MACv2-SP parametrisation and our PRP approach, we calculate instantaneous aerosol radiative forcing and derive annual means. Notably, in 2005, European, North American, and South Asian sources exhibit similar emission levels (refer to Table 6 in Stevens et al. [2017]), yet significant differences in direct effect efficiency are observed (as shown in Response Table 1). By normalizing the forcing by the respective AOD values (accounting for implicit regional processes), we observe a reduction in the regional efficiency spread. Furthermore, when analysing clear-sky aerosol forcing while accounting for regional cloud-masking effects, the spread is reduced further.

    Through our analysis of 2005 forcing and investigation of regional forcing disparities, we infer that a shift in emission patterns could potentially lead to a decoupling between global emissions and direct effect. Our study demonstrates this by the use of parametrised aerosol forcing and PRP approach to effectively distinguish between direct and indirect effects within MPI-ESM1.2. We note that a decoupling between direct effect and emissions does not necessarily imply a strong decoupling in total aerosol forcing, as the indirect effect is usually dominant in ESM [Fiedler et al., 2023] and more consistent with emissions (see Figure 1 and 2b). Such decomposition between direct and indirect effects is not as straightforward in models that use interactive aerosol modules. This complicates a direct comparison with models that explicitly represent aerosol processes. Acknowledging the limitations of our methodology, we constrain the focus of our study to radiative transfer processes in ESM and show that MACv2-SP is a valuable tool in this context.

2. We would like to address the comparison of the present-day aerosol direct effect magnitude

| Source region | Emissions | AOD | ADE | ADE/E | ADE/AOD |
|---|---|---|---|---|---|
| Europe | 16.41 | 2.79 | -0.012 | -0.72 | -4.22 |
| North America | 17.45 | 1.16 | -0.025 | -1.46 | -22.0 |
| East Asia | 37.36 | 4.26 | -0.086 | -2.30 | -20.18 |
| South Asia | 17.17 | 4.74 | -0.152 | -8.85 | -32.06 |
| North Asia | 1.70 | 0.55 | -0.005 | -3.09 | -9.54 |
| North Africa | 4.88 | 0.24 | -0.003 | -0.63 | -12.64 |
| South America | 4.15 | 0.45 | -0.012 | -2.83 | -26.07 |
| Maritime Continent | 3.35 | 1.38 | -0.003 | -0.80 | -1.94 |
| Australia | 1.57 | 0.56 | -0.026 | -16.73 | -47.01 |

Response Table 1: Aerosol direct effect efficiencies per source region in 2005. Emissions are Equivalent SO2 in Tg SO2, AOD [$10^{-3}$], Aerosol Direct Effect (ADE) in [$\mathrm{Wm^{-2}}$], ADE/E in [$10^{-3}\ \mathrm{Wm^{-2}}$] per emission unit, ADE/AOD in [$\mathrm{Wm^{-2}}$] per AOD unit

to the CMIP6 model mean. Referring to AR6, chapter 7, Table 7.6 [Forster et al., 2021], the Direct Effect CMIP6 average and 5-95% confidence range is $-0.25 \pm 0.40\ \mathrm{Wm^{-2}}$, Bellouin et al. [2020] reports a present day Direct Effect ranging from -0.37 to -0.12 $\mathrm{Wm^{-2}}$, whereas our study reports -0.324 $\mathrm{Wm^{-2}}$ (Figure 3.c). For the total aerosol radiative forcing, AR6 reports the CMIP6 average and 5-95% confidence range of $-1.11 \pm 0.38\ \mathrm{Wm^{-2}}$, Bellouin et al. [2020] report a present day total aerosol radiative forcing of -2.0 to 0.4 $\mathrm{Wm^{-2}}$ with a 90% likelihood, whereas our calculations stands at -0.76 $\mathrm{Wm^{-2}}$ (Figure 3.a). Other recent studies of aerosols radiative effects in CMIP6, such as Fiedler et al. [2023] and Smith et al. [2020], report a present day aerosol Effective Radiative Forcing ranging from -1.47 to -0.59 $\mathrm{Wm^{-2}}$ and from -1.37 to -0.63 $\mathrm{Wm^{-2}}$ respectively. Results from our PRP calculations align with other studies using different methods (e.g. Mauritsen et al. [2019], Fiedler et al. [2017]). Specifically, our results fall within the ranges cited above, particularly those from the AR6 assessment and Bellouin et al. [2020] study. This indicates that the magnitude of the present-day aerosol direct radiative forcing estimated in our study is not larger than the CMIP model mean and is consistent with the assessment provided in the AR6 report. It is important to clarify that MACv2-SP considers SO2, NH3, and BC as precursors. All emissions are presented in SO2 equivalent units, accounting for the respective contributions of these precursors. We acknowledge the need for clearer explanations in both the manuscript and figure captions about this.

3. We acknowledge your confusion regarding the discussion on SSA and biomass aerosols. While we extensively reference Stevens et al. [2017], we recognize the need for clearer information on the design of MACv2-SP plumes. We thus intend to clarify the effect of SSA in the manuscript as follow. Each plume represents anthropogenic aerosol emissions, accouting for SO2, NH3 and BC. The industrial plumes (Europe, North America, East and South Asia) emphasise the industrial aerosols, while the biomass plumes (North and South Central Africa, South America, Maritime Continental and Australia) emphasise aerosols resulting from anthropogenic biomass burning. This distinction is essential to reflect the dominance of specific aerosol species in each plume type. SSA values are uniformly set for both types of plumes based on measurements, as validated by Stevens [2015] to effectively capture temporal trends and regional patterns when compared to models integrating more complex aerosol processes.

4. The aerosol forcing in MPI-ESM1.2 falls within the mid CMIP6 range. In the paragraph starting on line 200 we refer to other studies that did extensive intercomparison between

CMIP6 models (such as Fiedler et al. [2023]); this type of comparison is outside the scope of the present study. As mentioned in Section 2, MPI-ESM1.2 successfully represents the historical surface temperature evolution, in particular it closely matches temperature records during the period 1950-1980 ("dampened warming" period), where the wider spread in CMIP6 models is observed [Mauritsen and Roeckner, 2020, Flynn and Mauritsen, 2020]. This makes us believe that MACv2-SP representation of aerosols is reasonable.

In summary, here is what we intend to include in the revised version of the manuscript:

- Scatter points on Figure 2 and 4 to highlight 2005 values (as presented here in Response Table 1), as well as more details on the temporal consistency of the plume efficiencies.

- A more exhaustive description of the MACv2-SP aerosol representation in the method Section, especially summarizing Section 4 on Time-varying forcing in Stevens et al. [2017].

- Clearer description of the effect of SSA and of the SO2-equivalent emission unit.

**References**

Bjorn Stevens, Stephanie Fiedler, Stefan Kinne, Karsten Peters, Sebastian Rast, Jobst Müsse, Steven J. Smith, and Thorsten Mauritsen. MACv2-SP: a parameterization of anthropogenic aerosol optical properties and an associated Twomey effect for use in CMIP6. *Geoscientific Model Development*, 2017. doi: 10.5194/gmd-10-433-2017.

Bjorn Stevens. Rethinking the lower bound on aerosol radiative forcing. *Journal of Climate*, 28(12): 4794–4819, jun 2015. doi: 10.1175/jcli-d-14-00656.1.

Stephanie Fiedler, Twan van Noije, Christopher J. Smith, Olivier Boucher, Jean-Louis Dufresne, Alf Kirkevåg, Dirk Olivié, Rovina Pinto, Thomas Reerink, Adriana Sima, and Michael Schulz. Historical Changes and Reasons for Model Differences in Anthropogenic Aerosol Forcing in CMIP6. *Geophysical Research Letters*, 50(15), aug 2023. doi: 10.1029/2023gl104848.

P. Forster, T. Storelvmo, K. Armour, W. Collins, J.-L. Dufresne, D. Frame, D.J. Lunt, T. Mauritsen, M.D. Palmer, M. Watanabe, M. Wild, and H. Zhang. The earth's energy budget, climate feedbacks, and climate sensitivity. *Climate Change 2021: The Physical Science Basis. Contribution of Working Group I to the Sixth Assessment Report of the Intergovernmental Panel on Climate Change [Masson-Delmotte, V., P. Zhai, A. Pirani, S.L. Connors, C. Péan, S. Berger, N. Caud, Y. Chen, L. Goldfarb, M.I. Gomis, M. Huang, K. Leitzell, E. Lonnoy, J.B.R. Matthews, T.K. Maycock, T. Waterfield, O. Yelekçi, R. Yu, and B. Zhou (eds.)].*, page 923–1054, 2021. doi: 10.1017/9781009157896.009.

N. Bellouin, J. Quaas, E. Gryspeerdt, S. Kinne, P. Stier, D. Watson-Parris, O. Boucher, K. S. Carslaw, M. Christensen, A.-L. Daniau, J.-L. Dufresne, G. Feingold, S. Fiedler, P. Forster, A. Gettelman, J. M. Haywood, U. Lohmann, F. Malavelle, T. Mauritsen, D. T. McCoy, G. Myhre, J. Mülmenstädt, D. Neubauer, A. Possner, M. Rugenstein, Y. Sato, M. Schulz, S. E. Schwartz, O. Sourdeval, T. Storelvmo, V. Toll, D. Winker, and B. Stevens. Bounding Global Aerosol Radiative Forcing of Climate Change. *Reviews of Geophysics*, 2020. doi: https://doi.org/10.1029/2019RG000660. URL https://agupubs.onlinelibrary.wiley.com/doi/abs/10.1029/2019RG000660.

Christopher J. Smith, Ryan J. Kramer, Gunnar Myhre, Kari Alterskjær, William Collins, Adriana Sima, Olivier Boucher, Jean-Louis Dufresne, Pierre Nabat, Martine Michou, Seiji Yukimoto, Jason Cole, David Paynter, Hideo Shiogama, Fiona M. O'Connor, Eddy Robertson, Andy Wiltshire, Timothy Andrews, Cécile Hannay, Ron Miller, Larissa Nazarenko, Alf Kirkevåg, Dirk Olivié, Stephanie Fiedler, Anna Lewinschal, Chloe Mackallah, Martin Dix, Robert Pincus, and Piers M. Forster. Effective radiative forcing and adjustments in CMIP6 models. *Atmospheric Chemistry and Physics*, 20(16):9591–9618, aug 2020. doi: 10.5194/acp-20-9591-2020.

Thorsten Mauritsen, Jürgen Bader, Tobias Becker, Jörg Behrens, Matthias Bittner, Renate Brokopf, Victor Brovkin, Martin Claussen, Traute Crueger, Monika Esch, Irina Fast, Stephanie Fiedler, Dagmar Fläschner, Veronika Gayler, Marco Giorgetta, Daniel S. Goll, Helmuth Haak, Stefan Hagemann, Christopher Hedemann, Cathy Hohenegger, Tatiana Ilyina, Thomas Jahns, Diego Jiménez de-la Cuesta, Johann Jungclaus, Thomas Kleinen, Silvia Kloster, Daniela Kracher, Stefan Kinne, Deike Kleberg, Gitta Lasslop, Luis Kornblueh, Jochem Marotzke, Daniela Matei, Katharina Meraner, Uwe Mikolajewicz, Kameswarrao Modali, Benjamin Möbis, Wolfgang A. Müller, Julia E. M. S. Nabel, Christine C. W. Nam, Dirk Notz, Sarah-Sylvia Nyawira, Hanna Paulsen, Karsten Peters, Robert Pincus, Holger Pohlmann, Julia Pongratz, Max Popp, Thomas Jürgen Raddatz, Sebastian Rast, Rene Redler, Christian H. Reick, Tim Rohrschneider, Vera Schemann, Hauke Schmidt, Reiner Schnur, Uwe Schulzweida, Katharina D. Six, Lukas Stein, Irene Stemmler, Bjorn Stevens, Jin-Song Storch, Fangxing Tian, Aiko Voigt, Philipp Vrese, Karl-Hermann Wieners, Stiig Wilkenskjeld, Alexander Winkler, and Erich Roeckner. Developments in the MPI-M Earth System Model version 1.2 (MPI-ESM1.2) and Its Response to Increasing CO2. *Journal of Advances in Modeling Earth Systems*, 11(4):998–1038, apr 2019. doi: 10.1029/2018ms001400.

S. Fiedler, B. Stevens, and T. Mauritsen. On the sensitivity of anthropogenic aerosol forcing to model-internal variability and parameterizing a Twomey effect. *Journal of Advances in Modeling Earth Systems*, 9(2):1325–1341, jun 2017. doi: 10.1002/2017ms000932.

Thorsten Mauritsen and Erich Roeckner. Tuning the MPI-ESM1.2 Global Climate Model to Improve the Match With Instrumental Record Warming by Lowering Its Climate Sensitivity. *Journal of Advances in Modeling Earth Systems*, 12(5), may 2020. doi: 10.1029/2019ms002037.

Clare Marie Flynn and Thorsten Mauritsen. On the climate sensitivity and historical warming evolution in recent coupled model ensembles. *Atmospheric Chemistry and Physics*, 2020. doi: 10.5194/acp-20-7829-2020.

---

## Author Response (AR2)

**General remarks on the reviews of the manuscript egusphere-2024-224**

We appreciate the valuable feedback and constructive comments from both reviewers, which will contribute to clarifying some aspects of the manuscript. Both reviewers have concerns about the aerosol parametrisation choices for representing the historical evolution of aerosol forcing. In our responses, we provide further description of the MACv2-SP aerosol parametrisation implemented in MPI-ESM1.2 and specific results of the year 2005 in which aerosol climatology is directly applied. We also include comparisons with studies suggested by the reviewers that support our results and conclusions.

Below, we respond point by point to both reviewers' comments, and provide a list of relevant changes in the manuscript.

**Anonymous Referee #1**

My understanding of the main objective of this manuscript is to explain why the global aerosol direct radiative forcing can exhibit a different trend than aerosol emissions in some climate models. The authors use the MPI-ESM1.2 model to quantify the historical direct, indirect and clear-sky aerosol forcings, and they find that the efficiency of aerosol direct radiative forcing is enhanced in the low-latitude regions such as the South and East Asia, where cloud masking is less and aerosol residence time is longer, compared to mid-latitude regions like Europe and North America. The work is interesting and falls within the journal's scope, but there are major concerns on the methodology employed in the current manuscript.

Reply: Thank you for your valuable comments. This study indeed aims to explore the potential decoupling between aerosol emissions and their direct effects. We acknowledge your concerns regarding the modelling approach employed to estimate historical aerosol forcing in MPI-ESM1.2 through the use of the MACv2-SP parametrization. In this response, we aim to provide clarity on our methodology by providing a description of the MACv2-SP parametrisation and specific results from our study.

1. The modeling approach to estimating the historical aerosol direct and indirect forcing from emissions used in MPI-ESM1.2 for this study is different from many CMIP6 models that represent aerosol effects through more explicit aerosol-cloud-radiation processes. This raises the concern of accuracy and consistency issues of the present study. The emissions of major anthropogenic species such as sulfur compounds and black carbon have been changing differently since 1980s across major source regions. However, the prescribed aerosol optical properties in the MPI-ESM1.2 simulations are based on measurements for 2005. I don't think the year-2005 aerosol properties are representative of the past few decades, especially for this research focused on the time evolution of aerosol forcing. Given that the key conclusion on the impact of changes in the aerosol residence time between mid-latitude and lower-latitude emission regions, I wonder whether the assumptions and model representation of aerosol optical properties reflect such changes. How large is the uncertainty in aerosol forcing comparing to the magnitude of direct and indirect forcing?

   Reply: MACv2-SP combines ground-based measurements of a 2005 aerosol climatology with emission estimates to represent historical changes in aerosol direct and indirect effects. Emissions estimates of SO2, NH3 as well as BC (CEDS) are accounted for in the scaling of the 2005 climatology (See Stevens et al. [2017]: Section 4: 'Time-varying forcing', Table 5,6, Figure 9,10). By weighting the radiative properties of these three compounds with their respective emission estimates, the changes in aerosol properties in time and space are represented. We acknowledge the limitation of this representation, which only takes into account these three (major) species and does not include interactive processes with the atmosphere. Nevertheless, Stevens [2015] argues that this representation of emissions successfully capture the main features seen in more complex models, both in terms of global signal and regional patterns. We intend to clarify and elaborate the representation of the time-varying forcing in the manuscript. While aerosol removal processes are not explicitly represented in MPI-ESM1.2 with MACv2-SP, they are recorded in the in-situ measurements and thus included implicitly in the MACv2-SP representation, and we argue that this provides a sufficient representation of aerosol forcing in the context of this study.

While we acknowledge your concern regarding the time-variation of aerosol forcing, we intend to provide specific results for the reference year of 2005, since it directly applies the measured aerosol climatology. Utilizing the MACv2-SP parametrisation and our PRP approach, we calculate instantaneous aerosol radiative forcing and derive annual means. Notably, in 2005, European, North American, and South Asian sources exhibit similar emission levels (refer to Table 6 in Stevens et al. [2017]), yet significant differences in direct effect efficiency are observed (as shown in Response Table 1). By normalizing the forcing by the respective AOD values (accounting for implicit regional processes), we observe a reduction in the regional efficiency spread. Furthermore, when analysing clear-sky aerosol forcing while accounting for regional cloud-masking effects, the spread is reduced further.

| Source region | Emissions | AOD | ADE | ADE/E | ADE/AOD |
|---|---|---|---|---|---|
| Europe | 16.41 | 2.79 | -0.012 | -0.72 | -4.22 |
| North America | 17.45 | 1.16 | -0.025 | -1.46 | -22.0 |
| East Asia | 37.36 | 4.26 | -0.086 | -2.30 | -20.18 |
| South Asia | 17.17 | 4.74 | -0.152 | -8.85 | -32.06 |
| North Asia | 1.70 | 0.55 | -0.005 | -3.09 | -9.54 |
| North Africa | 4.88 | 0.24 | -0.003 | -0.63 | -12.64 |
| South America | 4.15 | 0.45 | -0.012 | -2.83 | -26.07 |
| Maritime Continent | 3.35 | 1.38 | -0.003 | -0.80 | -1.94 |
| Australia | 1.57 | 0.56 | -0.026 | -16.73 | -47.01 |

Response Table 1: Aerosol direct effect efficiencies per source region in 2005. Emissions are Equivalent SO2 in Tg SO2, AOD $[10^{-3}]$, Aerosol Direct Effect (ADE) in $[Wm^{-2}]$, ADE/E in $[10^{-3} \ Wm^{-2}]$ per emission unit, ADE/AOD in $[Wm^{-2}]$ per AOD unit

Through our analysis of 2005 forcing and investigation of regional forcing disparities, we infer that a shift in emission patterns could potentially lead to a decoupling between global emissions and direct effect. Our study demonstrates this by the use of parametrised aerosol forcing and PRP approach to effectively distinguish between direct and indirect effects within MPI-ESM1.2. We note that a decoupling between direct effect and emissions does not necessarily imply a strong decoupling in total aerosol forcing, as the indirect effect is usually dominant in ESMs [Fiedler et al., 2023] and more consistent with emissions (see Figure 1 and 2b). Such decomposition between direct and indirect effects is not as straightforward in models that use

interactive aerosol modules. This complicates a direct comparison with models that explicitly represent aerosol processes. Acknowledging the limitations of our methodology, we constrain the focus of our study to radiative transfer processes in ESM and show that MACv2-SP is a valuable tool in this context.

2. The magnitude of present-day aerosol direct radiative forcing is larger than the CMIP model mean (Bellouin et al., 2020) and in the AR6 assessment. Is it because only sulfur aerosol is considered in the MPI-ESM1.2 model estimates? What other anthropogenic aerosol components (or precursor gases) are considered in the simulations? Figure 1 highlights the global anthropogenic SO2 emissions trend, but according to Hoesly et al. (2018), many other important components (e.g., BC, OC, NH3, NOx) had global increasing trends and very different regional trends in the past few decades. BC and OC are particularly important in aerosol direct forcing over South and East Asia, where this study emphasizes an increase in the sulfur emissions and direct forcing. Please explicitly evaluate the impact of BC and OC changes on the direct forcing trends.

Reply: We would like to address the comparison of the present-day aerosol direct effect magnitude to the CMIP6 model mean. Referring to AR6, chapter 7, Table 7.6 [Forster et al., 2021], the Direct Effect CMIP6 average and 5-95% confidence range is $-0.25 \pm 0.40$ Wm$^{-2}$, Bellouin et al. [2020] reports a present day Direct Effect ranging from -0.37 to -0.12 Wm$^{-2}$, whereas our study reports -0.324 Wm$^{-2}$ (Figure 3.c). For the total aerosol radiative forcing, AR6 reports the CMIP6 average and 5-95% confidence range of $-1.11 \pm 0.38$ Wm$^{-2}$, Bellouin et al. [2020] report a present day total aerosol radiative forcing of -2.0 to 0.4 Wm$^{-2}$ with a 90% likelihood, whereas our calculations stands at -0.76 Wm$^{-2}$ (Figure 3.a). Other recent studies of aerosols radiative effects in CMIP6, such as Fiedler et al. [2023] and Smith et al. [2020], report a present day aerosol Effective Radiative Forcing ranging from -1.47 to -0.59 Wm$^{-2}$ and from -1.37 to -0.63 Wm$^{-2}$ respectively. Results from our PRP calculations in MPI-ESM1.2 align with other studies using different methods (e.g. Mauritsen et al. [2019], Fiedler et al. [2017]). Specifically, our results fall within the ranges cited above, particularly those from the AR6 assessment and Bellouin et al. [2020] study. This indicates that the magnitude of the present-day aerosol direct radiative forcing estimated in our study is not larger than the CMIP model mean and is consistent with the assessment provided in the AR6 report. It is important to clarify that MACv2-SP considers SO2, NH3, and BC as precursors. All emissions are presented in SO2 equivalent units, accounting for the respective contributions of these precursors. We acknowledge the need for clearer explanations in both the manuscript and figure captions about this.

3. The discussion of aerosol SSA got me confused, as there is not sufficient information on the other aerosol species and why biomass aerosol is relevant to the focus of anthropogenic forcing of this study. I also wonder whether the SSA is set to a spatiotemporally uniform value in the simulations, which comes back to my major concern on the methodology.

Reply: We acknowledge your confusion regarding the discussion on SSA and biomass aerosols. While we extensively reference Stevens et al. [2017], we recognize the need for clearer information on the design of MACv2-SP plumes. We thus intend to clarify the effect of SSA in the manuscript as follow. Each plume represents anthropogenic aerosol emissions, accouting for SO2, NH3 and BC. The industrial plumes (Europe, North America, East and South Asia) emphasise the industrial aerosols, while the biomass plumes (North and South Central Africa, South America, Maritime Continental and Australia) emphasise aerosols resulting from anthropogenic biomass burning. This distinction is essential to reflect the dominance of specific

aerosol species in each plume type. SSA values are uniformly set for both types of plumes based on measurements, as validated by Stevens [2015] to effectively capture temporal trends and regional patterns when compared to models integrating more complex aerosol processes.

4. Although I don't expect the MPI-ESM1.2 results to be similar to other CMIP6 models, it would be nice to see how comparable the aerosol forcing estimates of this study to other models or observational analysis.

   Reply: The aerosol forcing in MPI-ESM1.2 falls within the mid CMIP6 range. In the paragraph starting on line 200 we refer to other studies that did extensive intercomparison between CMIP6 models (such as Fiedler et al. [2023]); this type of comparison is outside the scope of the present study. As mentioned in Section 2, MPI-ESM1.2 successfully represents the historical surface temperature evolution, in particular it closely matches temperature records during the period 1950-1980 ("dampened warming" period), where the wider spread in CMIP6 models is observed [Mauritsen and Roeckner, 2020, Flynn and Mauritsen, 2020]. This makes us believe that MACv2-SP representation of aerosols is reasonable.

**Anonymous Referee #2**

This short study uses the MPI-ESM climate model to suggest that aerosol direct radiative forcing has not followed the same decreasing trend as aerosol emissions. The authors explain that decoupling by an increase in aerosol radiative forcing efficiency (radiative forcing per unit emitted mass or per unit aerosol optical depth) due to a regional shift in aerosol distributions.

It is nice to see the possibility of a change in aerosol direct radiative forcing efficiency from changes in emission regions. That possibility has been mentioned in the past, including in the conclusion of the Bellouin et al. (2020) review paper cited by the authors, but I am not aware of a publication dedicated to the subject. The other results discussed by the authors (reduction in aerosol direct radiative forcing due to cloudiness, aerosol optical depth is more correlated than emissions with direct radiative forcing) have been discussed several times, especially in AeroCom papers, but they provide the context needed to explain the main result.

However, the study needs to go deeper when explaining the reasons for the change in radiative forcing efficiency, as I comment below. For this reason, I recommend additional analyses to clarify the drivers of aerosol direct radiative forcing efficiency in MPI-ESM.

Reply: Thank you for your insightful comments on our manuscript. We appreciate your attention to detail and would like to address the concerns you raised regarding the radiative forcing efficiency within the MACv2-SP aerosol prescription. We acknowledge the critical role of parametrisation choices in determining aerosol radiative forcing as well as the limitations of this method, and understand your point about the inherent influence of these choices. It is important to note that MACv2-SP was specifically developed to capture aerosol radiative effects on climate using ground-based and satellite measurements as well as emission inventories of key precursors such as SO2, NH3, and BC. In addition, MACv2-SP has been designed to be lightweight and easy to use in different climate models in the CMIP6 framework. Furthermore, Stevens [2015] argues that this representation successfully replicates key features observed in more complex models, both globally and regionally. To address your comments effectively, we will provide additional details on the construction of MACv2-SP to offer further clarity on our analysis. We also recognize the importance of

a thorough understanding of Stevens et al. [2017] for contextualizing our work, particularly Section 4: "Time-varying forcing".

Main comments:

The headline result that aerosol direct forcing is decoupled from aerosol emissions because of a change in radiative forcing efficiency is in many ways "built-in" the MAC-v2SP aerosol prescription used in MPI-ESM. The shapes and slopes of the curves shown on Figure 2 are a consequence of the choices made in MAC-v2SP in terms of aerosol plume properties and relative change in cloud droplet number. So the sensitivity to those choices needs to be explored more critically, specifically:

- There is a disconnect in the study between SSA and direct radiative forcing efficiencies that I do not understand. In Figure 2 and 4, why is direct radiative forcing efficiency in South Asia so strong? South Asia aerosols are known to be strongly absorbing, so I would have expected their direct radiative forcing to be less negative for a given AOD compared to regions dominated with more scattering aerosols. Moreover, the suggestion in section 3.5 that SSA does not significantly drive direct radiative forcing efficiency seems to go against AeroCom findings (e.g., the large differences in normalised radiative forcing shown between different aerosol types in the Tables of Myhre et al. (2013) https://doi.org/10.5194/acp-13-1853-2013 ) and I do not understand why. Perhaps MACv2-SP aerosols are too dominated by sulfate?

  Reply: We understand your concern about the strength of the South Asian plume. MACv2-SP uses CEDS emission inventories for scaling the forcing, including SO2, NH3 and BC for all plumes. Both reflective and absorbing aerosols are considered. MACv2-SP has been designed to fit climatological values of aerosol properties at the year 2005. "High-quality data by ground-based sun-photometer" are merged onto "global model background maps from AeroCom" [Stevens et al., 2017]. Different aerosol properties are considered: AOD at 440, 550 and 870 nm, absorbing AOD at 550 nm and coarse and fine-mode aerosol particles [Stevens et al., 2017]. Using this aerosol climatology retrieved from direct measurements, our calculation indicate a strong direct effect efficiency from South Asia, both against emissions and AOD in 2005 (see Response Table 1 in response to Referee #1). Since that the year 2005 applies aerosol climatology from measurements and AeroCom, we are uncertain why South Asian aerosols would be expected to be strongly absorbing.

  Furthermore, there must have been a misunderstanding about Section 3.5. We do suggest that SSA significantly drives direct effect forcing efficiency (line 174-175: "a new simulation where the SSA was set to the same value for all sources, substantially reducing the spread"). SSA does have an influence on the forcing, which explains the difference between plumes dominated by industrial aerosols and plumes dominated by biomass burning aerosols. But we emphasize that since the recent increases in biomass burning in the Southern Hemisphere have a minor contribution to the total aerosol direct effect, they have a minimal impact on the global forcing-to-emission decoupling.

- The link between residence times (different removal rates depending on region) discussed in Section 3.4 and MAC-v2SP is unclear. MAC-v2SP is a combination of global aerosol modelling outputs and satellite retrievals, so it may implicitly account for different residence times, but that would not influence radiative forcing efficiencies, since MACv2-SP simply scales plumes up and down with emission rates. Unless that Section 3.4 discussion is of a speculative nature?

  Reply: Emission rates are used to scale the 2005 aerosol climatology derived from measurements and AerCom [Stevens et al., 2017]. Analysis of the year 2005, where the measured climatology is applied (without any scaling), already suggests a strong regional dependence of the direct effect (see Response Table 1). Figure 2.c suggests that the difference between regions is in large part explain by cloud masking. Furthermore, Figure 4.a shows that the spread is reduced when considering AOD instead of emissions. To explain this, we do speculate that some aerosol processes are implicitly recorded in the instrumental measurements, such as aerosol removal. We do not completely understand your concern about the influence of residence times on radiative forcing, since we observe this discrepancy between emissions and AOD in 2005. Longer residence time results in greater measured AOD per unit of emission, resulting in greater forcing efficiency per unit of emission. Since the forcing is scaled with emissions inventories, the discrepancy is spread over time. Response Table 1 highlights the emissions and forcing values for each plume in the year 2005. The efficiencies in the Table are directly calculated from 2005 for comparison with the ones calculated in the manuscript figures. While in the manuscript we calculated the efficiencies via linear regression both against emissions and AODs, we acknowledge that we should more explicitly state that these efficiencies are consistent with emissions and thus time. We intent to clarify this in the manuscript.

Other comments:

- Lines 21-23: Note that Forster et al. (2021), cited just a few sentences earlier, prefers the aerosol-radiation and aerosol-cloud terminology instead of direct and indirect.

  Reply: Both direct and indirect effect and aerosol-radiation (ari) and aerosol-cloud interactions (aci) appear in the literature, and we find that the direct and indirect effect terminology are more intuitive and connect better with the framework of radiative forcing as an instantaneous effect on the radiative balance and subsequent interactions and adjustments within the climate system.

- Line 29: Quaas et al. (2022) https://doi.org/10.5194/acp-22-12221-2022, and especially their Section 5, seems a very relevant reference here, and elsewhere in the paper as well.

  Reply: We appreciate that you highlight the relevance of Quaas et al. [2022]. Section 5 of their work emphasizes the close relationship between clear-sky solar ERFaer trends and trends in sulfate precursors, noting significant declines in major source regions from North America, Europe, and East Asia, alongside increases in India and surrounding regions. These findings align with our results across the 4 major industrial plumes and address your concerns regarding the strength of South Asia and the dominance of sulfate in MACv2-SP. Notably, their study supports our results by indicating that trends in sulfate precursors are driving aerosol forcing increases in regions like 'India and surrounding areas' (referred to as South Asia in MACv2-SP) over recent decades. We intend to include this comparison into the manuscript in Section 3.2.

- Line 73: "as a two-sided version". What does that mean?

  Reply: We systematically employ the two-sided PRP method described in Klocke et al. [2013]. To keep our text concise, we opted not to delve into a detailed explanation of the PRP methodology in this paper. The original forward PRP (Wetherald and Manabe [1988]) consists in sequentially substituting specific state variable fields from the current climate state into a reference state, while keeping all other variables at the reference level. This partial perturbation approach allows for assessing each variable's contribution to the total radiative forcing. However, the perturbation is sensitive to the state in which it is introduced [Colman and McAvaney, 1997] and de-correlating fields when substituting them introduces unintended perturbations

[Klocke et al., 2013]. The proposed method to partially address these approximations is to apply the partial radiative perturbation forward and backward. The backward perturbation consists in substituting the variable fields from the reference state into the current state. By averaging the results of forward and backward computations (two-sided approach), a more accurate estimation of forcing is achieved. We judge that a detailed description of this methodology would be too technical for the scope of this paper, thus we refer the reader to Klocke et al. [2013] and Colman and McAvaney [1997] for a more comprehensive description.

- Line 103: How are those "Tg of SO2 equivalent" calculated?

  Reply: As mentioned earlier, emissions in Tg of SO2 equivalent are calculated taking into account SO2, NH3 and BC emission inventories from CEDS. We acknowledge that we should more explicitly highlight this in the paper and intent to do so in the revised version, since we use SO2 equivalent as a unit in our main results. For a full description of SO2 equivalent calculation, the reader can refer to Stevens et al. [2017].

- Line 128: "absorption prevails in the presence of clouds" – only if the aerosols are above clouds.

  Reply: You are right that absorption prevails in the presence of clouds only if the aerosols are above clouds. We will mention this in section 3.3.

- Line 151-152: Do you use the latest CEDS emissions for the MACv2-SP plume scaling, dated 21 April 2021? There have been significant changes in aerosol emission trends, especially over China, that would affect the results presented here.

  Reply: We use the MACv2-SP version from Stevens et al. [2017] that integrates the CEDS emission estimates used for CMIP6 historical forcing input data. As mentioned previously, the MACv2-SP reference year 2005 uses instrumental measurements of the aerosol climatology. Since our results for the year 2005 already suggest a regional dependence of the aerosol efficiency and imply the decoupling, we do not think that more recent emission estimates would affect the results. The emission estimates are only used to scale the 2005 reference to represent the time-varying forcing. In the context of our study, this helps to clearly observe the decoupling in Figure 1, but our key results are the spatial representation with Figure 2,3 and 4.

  Additionally, the results from Quaas et al. [2022] uses these 2021 updated CEDS emission estimates. Using this new dataset they obtain the results discussed in our answer to your comment on Line 29. Since their results are consistent with our results, we believe that this update would have little effect on our results, and is unlikely to impact the outcomes of our research.

**List of changes in the manuscript**

- We divide the Method section in two part, "2.1 MPI-ESM1.2 and MACv2-SP" focuses only on the aerosol parametrisation in MPI-ESM1.2. while "2.2 Radiative Forcing Calculation" describes our approach to compute aerosol direct and indirect effects. In 2.1., further description of the MACv2-SP parametrisation from Stevens et al. [2017] to clarify the method for representing the time-variations in aerosol forcing, specifically mentioning the different aerosol precursors taken into account. We also clarify the [Tg of SO2 equivalent] unit used throughout our study. In 2.2, we added a brief description of "two-sided" PRP calculation, but we refer the reader to relevant publication for a more comprehensive description.

- In section 3.2., we included a comparison with Quaas et al. [2022], supporting that trends in clear-sky aerosol forcing is mainly driven by trends in sulfate precursors. We also added black crossed in Figure 2 which highlight the 2005 values and showing discrepancy in aerosol efficiency among regions when AeroCom climatology is applied.

- In section 3.3., we mention that the clouds moderates the effect of aerosols when clouds a situated "below aerosols".

- In section 3.4., we clarify that aerosol removal may implicitly be recorded in MACv2-SP.

- In 3.5., we clarify the distinction between industrial and biomass burning plumes through the SSA parameter and emphasise that aerosol properties influence the resulting forcing.

**References**

Bjorn Stevens, Stephanie Fiedler, Stefan Kinne, Karsten Peters, Sebastian Rast, Jobst Müsse, Steven J. Smith, and Thorsten Mauritsen. MACv2-SP: a parameterization of anthropogenic aerosol optical properties and an associated Twomey effect for use in CMIP6. *Geoscientific Model Development*, 2017. doi: 10.5194/gmd-10-433-2017.

Bjorn Stevens. Rethinking the lower bound on aerosol radiative forcing. *Journal of Climate*, 28(12): 4794–4819, jun 2015. doi: 10.1175/jcli-d-14-00656.1.

Stephanie Fiedler, Twan van Noije, Christopher J. Smith, Olivier Boucher, Jean-Louis Dufresne, Alf Kirkevåg, Dirk Olivié, Rovina Pinto, Thomas Reerink, Adriana Sima, and Michael Schulz. Historical Changes and Reasons for Model Differences in Anthropogenic Aerosol Forcing in CMIP6. *Geophysical Research Letters*, 50(15), aug 2023. doi: 10.1029/2023gl104848.

P. Forster, T. Storelvmo, K. Armour, W. Collins, J.-L. Dufresne, D. Frame, D.J. Lunt, T. Mauritsen, M.D. Palmer, M. Watanabe, M. Wild, and H. Zhang. The earth's energy budget, climate feedbacks, and climate sensitivity. *Climate Change 2021: The Physical Science Basis. Contribution of Working Group I to the Sixth Assessment Report of the Intergovernmental Panel on Climate Change [Masson-Delmotte, V., P. Zhai, A. Pirani, S.L. Connors, C. Péan, S. Berger, N. Caud, Y. Chen, L. Goldfarb, M.I. Gomis, M. Huang, K. Leitzell, E. Lonnoy, J.B.R. Matthews, T.K. Maycock, T. Waterfield, O. Yelekçi, R. Yu, and B. Zhou (eds.)].*, page 923–1054, 2021. doi: 10.1017/9781009157896.009.

N. Bellouin, J. Quaas, E. Gryspeerdt, S. Kinne, P. Stier, D. Watson-Parris, O. Boucher, K. S. Carslaw, M. Christensen, A.-L. Daniau, J.-L. Dufresne, G. Feingold, S. Fiedler, P. Forster, A. Gettelman, J. M. Haywood, U. Lohmann, F. Malavelle, T. Mauritsen, D. T. McCoy, G. Myhre, J. Mülmenstädt, D. Neubauer, A. Possner, M. Rugenstein, Y. Sato, M. Schulz, S. E. Schwartz, O. Sourdeval, T. Storelvmo, V. Toll, D. Winker, and B. Stevens. Bounding Global Aerosol Radiative Forcing of Climate Change. *Reviews of Geophysics*, 2020. doi: https://doi.org/10.1029/2019RG000660. URL https://agupubs.onlinelibrary.wiley.com/doi/abs/10.1029/2019RG000660.

Christopher J. Smith, Ryan J. Kramer, Gunnar Myhre, Kari Alterskjær, William Collins, Adriana Sima, Olivier Boucher, Jean-Louis Dufresne, Pierre Nabat, Martine Michou, Seiji Yukimoto, Jason Cole, David Paynter, Hideo Shiogama, Fiona M. O'Connor, Eddy Robertson, Andy Wiltshire, Timothy Andrews, Cécile Hannay, Ron Miller, Larissa Nazarenko, Alf Kirkevåg, Dirk Olivié,

Stephanie Fiedler, Anna Lewinschal, Chloe Mackallah, Martin Dix, Robert Pincus, and Piers M. Forster. Effective radiative forcing and adjustments in CMIP6 models. *Atmospheric Chemistry and Physics*, 20(16):9591–9618, aug 2020. doi: 10.5194/acp-20-9591-2020.

Thorsten Mauritsen, Jürgen Bader, Tobias Becker, Jörg Behrens, Matthias Bittner, Renate Brokopf, Victor Brovkin, Martin Claussen, Traute Crueger, Monika Esch, Irina Fast, Stephanie Fiedler, Dagmar Fläschner, Veronika Gayler, Marco Giorgetta, Daniel S. Goll, Helmuth Haak, Stefan Hagemann, Christopher Hedemann, Cathy Hohenegger, Tatiana Ilyina, Thomas Jahns, Diego Jiménéz de-la Cuesta, Johann Jungclaus, Thomas Kleinen, Silvia Kloster, Daniela Kracher, Stefan Kinne, Deike Kleberg, Gitta Lasslop, Luis Kornblueh, Jochem Marotzke, Daniela Matei, Katharina Meraner, Uwe Mikolajewicz, Kameswarrao Modali, Benjamin Möbis, Wolfgang A. Müller, Julia E. M. S. Nabel, Christine C. W. Nam, Dirk Notz, Sarah-Sylvia Nyawira, Hanna Paulsen, Karsten Peters, Robert Pincus, Holger Pohlmann, Julia Pongratz, Max Popp, Thomas Jürgen Raddatz, Sebastian Rast, Rene Redler, Christian H. Reick, Tim Rohrschneider, Vera Schemann, Hauke Schmidt, Reiner Schnur, Uwe Schulzweida, Katharina D. Six, Lukas Stein, Irene Stemmler, Bjorn Stevens, Jin-Song Storch, Fangxing Tian, Aiko Voigt, Philipp Vrese, Karl-Hermann Wieners, Stiig Wilkenskjeld, Alexander Winkler, and Erich Roeckner. Developments in the MPI-M Earth System Model version 1.2 (MPI-ESM1.2) and Its Response to Increasing CO2. *Journal of Advances in Modeling Earth Systems*, 11(4):998–1038, apr 2019. doi: 10.1029/2018ms001400.

S. Fiedler, B. Stevens, and T. Mauritsen. On the sensitivity of anthropogenic aerosol forcing to model-internal variability and parameterizing a Twomey effect. *Journal of Advances in Modeling Earth Systems*, 9(2):1325–1341, jun 2017. doi: 10.1002/2017ms000932.

Thorsten Mauritsen and Erich Roeckner. Tuning the MPI-ESM1.2 Global Climate Model to Improve the Match With Instrumental Record Warming by Lowering Its Climate Sensitivity. *Journal of Advances in Modeling Earth Systems*, 12(5), may 2020. doi: 10.1029/2019ms002037.

Clare Marie Flynn and Thorsten Mauritsen. On the climate sensitivity and historical warming evolution in recent coupled model ensembles. *Atmospheric Chemistry and Physics*, 2020. doi: 10.5194/acp-20-7829-2020.

Johannes Quaas, Hailing Jia, Chris Smith, Anna Lea Albright, Wenche Aas, Nicolas Bellouin, Olivier Boucher, Marie Doutriaux-Boucher, Piers M. Forster, Daniel Grosvenor, Stuart Jenkins, Zbigniew Klimont, Norman G. Loeb, Xiaoyan Ma, Vaishali Naik, Fabien Paulot, Philip Stier, Martin Wild, Gunnar Myhre, and Michael Schulz. Robust evidence for reversal of the trend in aerosol effective climate forcing. *Atmospheric Chemistry and Physics*, 22(18):12221–12239, September 2022. ISSN 1680-7324. doi: 10.5194/acp-22-12221-2022.

Daniel Klocke, Johannes Quaas, and Bjorn Stevens. Assessment of different metrics for physical climate feedbacks. *Climate Dynamics*, 2013. doi: 10.1007/s00382-013-1757-1.

R. T. Wetherald and S. Manabe. Cloud Feedback Processes in a General Circulation Model. *Journal of the Atmospheric Sciences*, 1988. doi: 10.1175/1520-0469(1988)045⟨1397:cfpiag⟩2.0.co;2.

R. A. Colman and B. J. McAvaney. A study of general circulation model climate feedbacks determined from perturbed sea surface temperature experiments. *Journal of Geophysical Research: Atmospheres*, 1997. doi: 10.1029/97jd00206.

---

## Author Response (AR3)

**General remarks on the reviews for minor revisions of the manuscript egusphere-2024-224**

We thank both referees for reviewing and supporting the revisions we made on the first version of the manuscript. We acknowledge that the representation of aerosol composition in MACv2-SP, as well as the 'built-in' effects of the parametrization, have to be clarified throughout the manuscript. Below, we respond point by point to the comments left by Referee #2, and provide a list of relevant changes in the manuscript.

**Anonymous Referee #2**

I thank the authors for considering my comments, and especially for clarifying how aerosol absorption is represented in MACv2-SP. I have read section 4 of Stevens et al. (2017), which confirms that changes in radiative efficiency due to changes in aerosol composition in the different regional plumes are not represented in MACV2-SP. That choice was made despite evidence of there having been changes in radiative efficiency due to changes in aerosol composition. Stevens et al. (2017) resolves that contradiction by saying "experiments to judge the magnitude of [aerosol composition] changes are warranted" (last paragraph of their Section 4).

Reply: Thank you very much for reading in detail section 4 of Stevens et al. [2017] and providing valuable external view on our work. Changes in aerosol composition are indeed not represented in MACv2-SP more than the fractions of $SO_2$ and $NH_3$. Stevens et al. [2017] suggest that the 'brightening of aerosols' from the reduction in BC fraction prior to 1970 (Figure 10) should results in an increase in efficiency, while the opposite trend after 1970 should imply a decrease in efficiency. The evolution of aerosol efficiency over time may be significantly influenced by changes in aerosol composition. However, these changes typically occur alongside variations in aerosol emissions, potentially limiting their impact on global efficiency. The increase in efficiency after 1980 in our findings appears to contradict with the observations mentioned above. We address this inconsistency in greater detail below, where we discuss temporal changes in aerosol composition and their implications for our results.

That built-in behaviour of MACv2-SP has clear implications for the present study. That lack of representation of changes driven by aerosol composition may let other drivers emerge, like cloud masking. So this MACv2-SP behaviour must be clearly documented in the paper: in the abstract, in the last paragraph of section 2.1, and in section 3.5, and in the conclusion, which could suggest a similar analysis in more complex aerosol-climate models.

Reply: We agree that the design choices made in MACv2-SP have implications for the results of our study. In this revised version, we further discuss the impact of not including changes in aerosol compositions within each plume.

Regarding cloud masking, our study focused on the observed trend in the direct effect of aerosols. We did not investigate the trend in the indirect effect, thus cloud masking falls outside the scope of this research. Nevertheless, our methodology provides relevant insights: By employing the PRP method, we effectively separated the direct and indirect effects of aerosols. This approach isolates the forcing from the direct effect, independently of any masking effects due to aerosol indirect effects. Notably, we found that the total aerosol forcing equals the sum of the isolated direct and indirect effects, indicating that the indirect effect does not mask the direct effect through enhanced cloud masking. It would be indeed valuable to investigate such effects in more complex aerosol-climate

models, provided these models can effectively dissociate direct and indirect effects similarly to our study with the PRP method.

There is also a point where the present paper seems to contradict Stevens et al. (2017). The authors write that in the revised section 2.1 that the plume scale factor applied to the reference year (2005) depends on SO2, NH3, and BC emissions. But Eq. 17 and Table 5 of Stevens et al. (2017) clearly say that only SO2 and NH3 are considered, not BC. The mention of BC in their Table 6 is only informative, although it suggests that accounting for BC would have been a good thing. As written in that paper, "Other factors, which do not correlate with regional NH3 and SO2 emissions" are not represented. So a decorrelation between BC and SO2 emissions (and the subsequent change in radiative forcing efficiency) is not possible in MACv2-SP. I am not sure to what extent that matters for this study – it depends on whether the decorrelation has happened in reality, especially in Southeast Asia. But the description of the method should make clear that BC is not involved in the scaling factors, and the implications of that choice should be discussed in sections 3.5 and the conclusion.

Reply: Thank you for noticing this contradiction. We clarified in our Method section that BC is not explicitly considered in the MACv2-SP. This is particularly relevant for South Asia, as this region has been driving the trend in total direct effect in recent decades (our results, Quaas et al. [2022], etc.). Inventories suggest that South Asia has maintained a consistent aerosol composition since 1970, with the BC fraction relative to 2005 levels remaining stable (approximately 1, as shown in Table 6, Stevens et al. [2017]). This consistency implies that aerosol efficiency in South Asia has likely remained stable over the past few decades. Therefore, including changes in aerosol composition over time, such as the BC fraction, would not significantly impact our results and would not influence the overall discrepancy between total direct effect and global emissions. Even though the aerosol efficiencies in European and North American regions have decreased since 1980, their contribution to the total efficiency is relatively small compared to that of Asian regions. Furthermore, emerging biomass burning plumes in recent decades have exhibited lower Single Scattering Albedo (SSA) values, representing an increase in absorbing aerosols and a resulting decrease in total efficiency. However, since the relative emissions from these biomass burning plumes are small, they do not counterbalance the increase in total efficiency driven by Asian plumes.

**List of changes in the manuscript**

We included the discussion on aerosol composition in the relevant sections as suggested by Referee #2:

- We specified in the abstract that MACv2-SP employs a constant regional direct effect efficiency.

- We clarified in Section 2.1 that BC is not included in the MACv2-SP aerosol composition for time-varying aerosol forcing and that changes in aerosol composition over time is not included. Furtheremore, we added that some of these global changes in aerosol composition are implied through the distinction between industrial and biomass burning plumes. We then emphasized that we consider these limitations in our analysis.

- In the Result section, 3.5, we discuss the implications of excluding changes in aerosol composition on our findings, as discussed in the answer to Referee #2.

- Finally, we summarized this discussion in the conclusion and suggest similar study in more explicit aerosol-climate models.

**References**

Bjorn Stevens, Stephanie Fiedler, Stefan Kinne, Karsten Peters, Sebastian Rast, Jobst Müsse, Steven J. Smith, and Thorsten Mauritsen. MACv2-SP: a parameterization of anthropogenic aerosol optical properties and an associated Twomey effect for use in CMIP6. *Geoscientific Model Development*, 2017. doi: 10.5194/gmd-10-433-2017.

Johannes Quaas, Hailing Jia, Chris Smith, Anna Lea Albright, Wenche Aas, Nicolas Bellouin, Olivier Boucher, Marie Doutriaux-Boucher, Piers M. Forster, Daniel Grosvenor, Stuart Jenkins, Zbigniew Klimont, Norman G. Loeb, Xiaoyan Ma, Vaishali Naik, Fabien Paulot, Philip Stier, Martin Wild, Gunnar Myhre, and Michael Schulz. Robust evidence for reversal of the trend in aerosol effective climate forcing. *Atmospheric Chemistry and Physics*, 22(18):12221–12239, September 2022. ISSN 1680-7324. doi: 10.5194/acp-22-12221-2022.